# Evaluation of the Effectiveness of Trail Repair Works Based on Three-Dimensional Monitoring around Mount Kurodake, Daisetsuzan National Park, Japan

Yusuke Kobayashi [1] and Teiji Watanabe [2,*]

1 Graduate School of Environmental Science, Hokkaido University, Hokkaido, Sapporo 060-0810, Japan; ykobayashi48@icloud.com
2 Faculty of Environmental Earth Science, Hokkaido University, Hokkaido, Sapporo 060-0810, Japan
* Correspondence: twata@ees.hokudai.ac.jp

**Abstract:** Many mountainous, protected areas, such as national parks worldwide, face trail erosion; stakeholders have made significant efforts to manage eroded trails. However, their effectiveness has not been evaluated. This study aimed to (1) create digital elevation models of an eroded trail using structure-from-motion and multi-view-stereo photogrammetry in Daisetsuzan National Park, northern Japan; (2) conduct a six-year monitoring of the trails repaired by volunteers to reveal trail surface changes; and (3) discuss the effectiveness of the repair works. Palm-fiber bags were used on the trail section to stop the movement of the eroded soil. The results of the three-dimensional analysis identified a certain effectiveness of the repair work during 2017–2022. However, the effectiveness lasted for only approximately three years and was not permanent. Therefore, regular maintenance is necessary to ensure trail sustainability. In addition, the soil erosion rate calculated using the maximum erosion depth has increased from 0.52 mm y$^{-1}$ (1923–1990) to 44.4 mm y$^{-1}$ (2013–2022), suggesting the need for frequent observations. Trail maintenance through a combination of monitoring and repair work is vital, and the role of hikers/trekkers is becoming increasingly important.

**Keywords:** trail erosion; trail management; trail monitoring; anthropogenic impact

## 1. Introduction

Hiking and trekking are popular recreational activities in the mountainous areas of Japan. According to the National Census, approximately seven million people are involved in mountain activities [1]. These mountain activities typically require formal trails. Trails are regarded as important infrastructure essential for hikers/trekkers to reach their destinations [2,3]. Protected area managers conserve the surrounding natural environment from human disturbance, for example, by concentrating trekker traffic on trails [4]. Trails must be properly managed to ensure their safe use by hikers/trekkers and the conservation of natural landscapes. Nevertheless, challenges related to trails are evident in many protected areas worldwide [5,6].

The impacts of trampling by hikers/trekkers on trails include degradation, such as trail erosion, trail widening, trail multiplication, and root exposure [7,8]. Trail degradation has a negative impact on valuable ecology, leading to the degradation and loss of vegetation [9–11]. It also worsens the aesthetic landscape and the quality of the recreational experience [12,13].

Trail science, which focuses on trail degradation and management, has primarily aimed at protected area management, and began in the 1970s [14,15]. The author of ref. [14] focused on expanding trail width and found that steeper trails tended to expand the width of Scottish hill paths.

Basic methods for measuring mountain trails were established in the 1980s [16,17]. The study by ref. [17] in Rocky Mountain National Park found that the degree of trail erosion

is owing to the interaction of geomorphic processes and meteorological events, and is governed by the topographic conditions of where the trail is located. For the measurement of trails, a census method classifies trail segments based on the degree of erosion, by recording the maximum width and depth of the trails along with the problems related to the trails [17,18]. This method is simple and is suitable for recording a wide range of mountain trails. The authors of ref. [18] measured the depth of erosion in Taiwan and combined the results with soil penetration tests to predict future erosion. The cross-sectional area measurement method has been most widely used in trail degradation research [16,19]. Tape or sticks are fixed horizontally on both sides of the trail edge and the depth to the surface of the eroded trail is measured at the same interval after a certain period. The eroded area can be easily calculated as the difference between the cross-sectional areas. The cross-sectional area measurement method is inexpensive and simple to use to understand the changes and characteristics of trails over time. Data obtained from monitoring using the cross-sectional area measurement is essential to protect park resources and the quality of visitor experiences [19]. The authors of ref. [20] proposed the concepts of maximum-incision post-trail construction and maximum-incision current tread to improve the cross-sectional area measurement based on the shape of the terrain and changes in trail use. They found that the amount of use was significantly related to the trail width and erosion.

The study by ref. [21] on El Portalet mountain trails in the eastern Iberian Peninsula developed a method to estimate the overall erosion volume by measuring only the width and maximum erosion depth. Monitoring is especially important for management decision making and the above methods have been introduced into trail science. Simple and reproducible methods are important for the management of protected areas, such as national parks. After severe erosion occurs, repairs become costly. Therefore, a simplified method is suitable for trail impact assessment, leading to the early detection of problems on trails. In addition, this simple and repeatable method will contribute to the development of a management system that involves not only researchers but also park managers and hikers/trekkers [18].

These traditional monitoring methods adopt a two-dimensional approach; hence, it is difficult to understand the entire trail or trail section spatially. Spatially quantifying erosion and sedimentation is time consuming, labor intensive, and difficult. The accurate and efficient measurement of trail erosion is considered the most difficult part of trail science [22]. The authors of ref. [23] pointed out the above problems and developed a method to analyze the spatial and temporal aspects of small changes on a trail surface. They created digital elevation models for southern Poland using a total station, and showed spatial changes over the study period. These results allowed the researchers and protected-area managers to calculate the erosion volume over the entire trail section.

With recent advancements in survey technology, new measurement methods have been developed. In particular, unmanned aerial vehicles (UAVs) began to be adopted in the 2010s, and a new photogrammetric survey method, that is, structure-from-motion and multi-view-stereo photogrammetry, has been applied to geomorphological studies [24–26], including trail science [27,28]. The authors of ref. [27] confirmed that structure-from-motion and multi-view-stereo photogrammetry with UAVs and a smartphone camera provided results similar to those of cross-sectional area measurement. Close-range photogrammetry, including pole photography, has mostly been used in geography [29]. The effectiveness of pole photography for topographical change detection has already been described in several papers [29–31]. Other studies using the latest technologies include (1) airborne LiDAR, which was used on the Appalachian Trail [32], and (2) dendrochronology [33,34].

Japan has 34 national parks, including many mountainous ones. Walking off the tread surface, designated as formal trails, is prohibited in Japan's national parks. Consequently, the impacts of hikers/trekkers are concentrated on the formal trails, resulting in erosion. There are many studies on the various challenges related to trails caused by the concentration of hikers/trekkers and heavy precipitation from the summer monsoon and winter snowfall in Japan, although most of the papers are written in Japanese [35–41].

Daisetsuzan National Park is one of the few mountainous national parks in the world where the cross-sectional area measurement method has been applied over the long-term (>10 years) (e.g., [42]). Studies of trails in high-elevation and/or cold climates similar to Daisetsuzan National Park have been conducted in the U.S., Iceland, Nepal, Greece, and Peru [43–49]. Alpine environments are sensitive to changes and impacts, such as from trail use. It is not only the number of visitors that determines the rate and magnitude of trail degradation, but also the relationship of the trail to the terrain and landform, ground characteristics, and weather conditions [48].

The Japanese Ministry of the Environment and local governments manage mountain trails in national parks; however, there is no official management or construction organization, owing to a lack of personnel and budget. Most trail repairs are voluntarily conducted by mountain lodge owners and local mountaineering associations. However, traditional voluntary trail management has faced difficulties owing to the aging of technical staff and volunteers in recent decades, and the deterioration of business conditions caused by COVID-19 over the past few years. Therefore, more hikers/trekkers are invited, as volunteers, to maintain and manage the trails and to be involved in repair work in Japan's mountain national parks, including Daisetsuzan National Park. Hikers/trekkers are highly motivated to participate in repair work because they can contribute to nature conservation through their work. Such initiatives have the potential to spread to more protected areas and other mountain trails in Japan as well as other parts of the world.

However, no studies have scientifically confirmed the effectiveness of these repair methods. The evaluation of management actions, such as repair work, is necessary for trail science [12]. Trail science has focused primarily on quantitative measurements of erosion, with little attention paid to the destination of the eroded sediment. However, with the advent of simple survey methods such as unmanned aerial vehicles, pole photography, and the structure-from-motion and multi-view-stereo photogrammetry, the above challenges may be solved. Although many monitoring studies of trail erosion have been conducted, this study is the first attempt to adopt a combined approach of long-term monitoring over a six-year period and an in situ observation study on the effectiveness of repair works. Clarifying the effectiveness of repair work could increase the motivation of voluntary participants. This study aimed to (1) create digital elevation models (DEMs) of eroded trails using structure-from-motion and multi-view-stereo photogrammetry, (2) conduct long-term monitoring of the trail repaired by volunteers to reveal trail surface changes, and (3) discuss the effectiveness of the repair works.

## 2. Materials and Methods

### 2.1. Study Area

#### 2.1.1. Daisetsuzan National Park

Daisetsuzan National Park was designated in 1934 as one of the oldest national parks in Japan. Daisetsuzan National Park has an area of 2267.64 km$^2$ and is the largest national park in the country. The approximate annual number of hikers/trekkers in the park was estimated to be 90,000 ± 10,000 in 2019 and 100,000 ± 10,000 in 2022 (unpublished data provided by the Ministry of the Environment). Thick snow covers the ground surfaces from early October to the end of June. Therefore, almost all trekkers are concentrated during the short period from July to September. No mountain biking or horseback riding is allowed on the trails in the alpine zones of Japan's national parks. In recent years, trail running has been gradually increasing [50], although park guidelines suggest no trail running in Daisetsuzan National Park.

Mount Daisetsuzan, which refers to the mountain massif including Mount Asahi-dake, the highest peak (2291 m) in Hokkaido (Figure 1), was formed by older (ca. 1.0–0.7 Ma) and younger (ca. 0.2 Ma to present) volcanic activity [51].

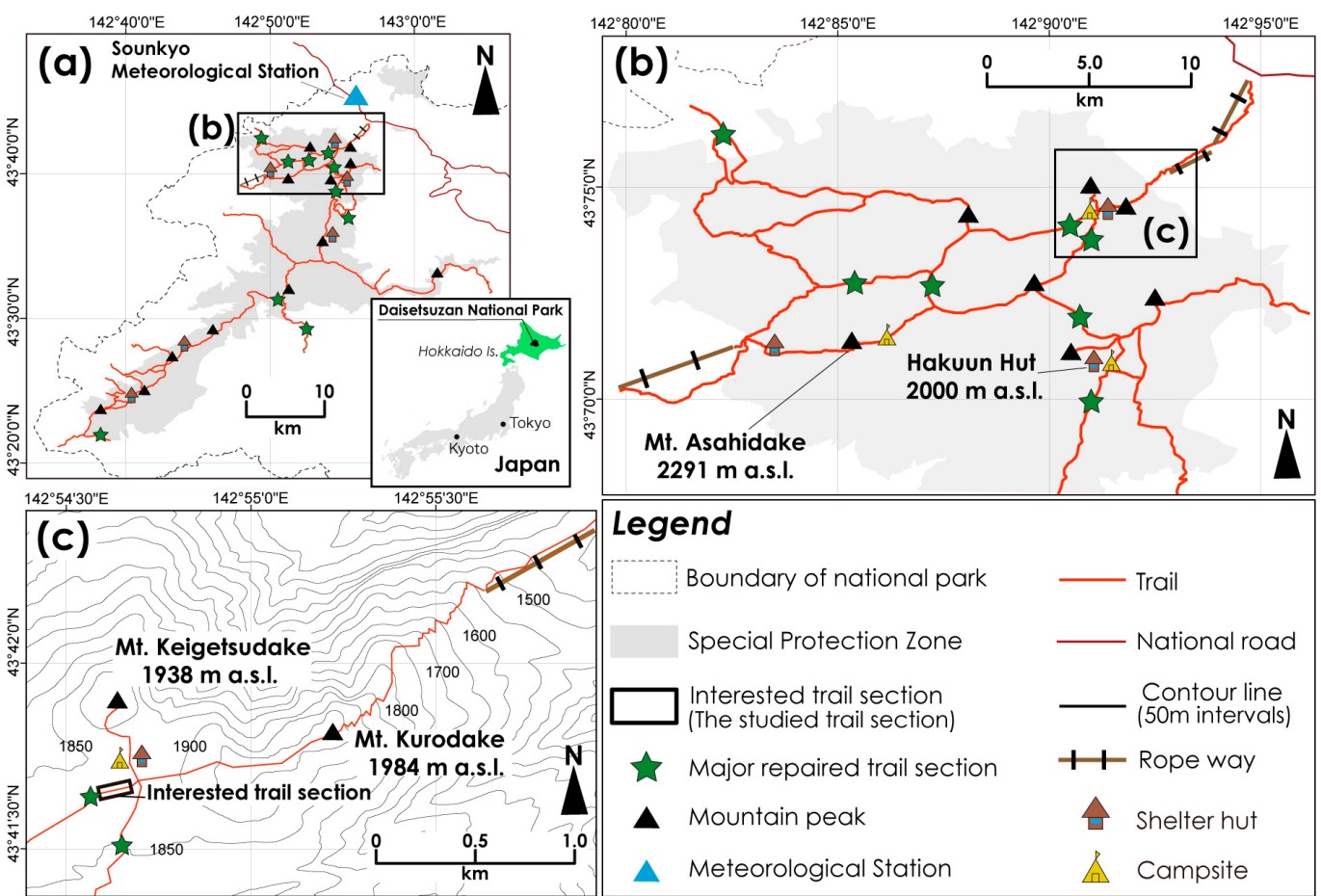

**Figure 1.** Study area: (**a**) major part of Daisetsuzan National Park; (**b**) northern part of the park; and (**c**) location of the studied trail section.

The mean annual air temperature recorded at Hakuun Hut (2000 m a.s.l.) was −3.8 °C, the lowest monthly air mean temperature in January was −21.3 °C, and the highest monthly mean air temperature in August was 13.9 °C in 1985 [52]. The area above 1850 m is widely covered by periglacial landforms, such as sorted polygons, sorted stripes, frost cracks, turf-banked terraces, earth hummocks, and palsas [53–55]. Permafrost also exists above 1700 m [56–58]. The average number of days of precipitation from 2010 to 2022 was 173 days per year [59].

2.1.2. Studied Trail Section

The total length of the mountain trails in the park area is approximately 300 km. The trails are classified into five levels based on the grading system (Daisetsuzan Grade): Grade 1 (short trails to enjoy the beautiful nature) to Grade 5 (trails with extremely challenging terrain). These five grades are determined by the degree of pristine nature, vulnerability of the natural environment, weather conditions, trail distance, presence of shelters and camping grounds, and access to the trailhead [60]. The trail passing through the study area was designated as Grade 3 (accessible by day hikes/trekkers with intermediate experience). The studied trail section (Figure 1b) is located in an alpine zone under the special protection zone designated by the Ministry of the Environment.

The studied trail section, with a length of 133 m, is located on flat terrain, called the *Kumono-daira*, to the southwest of Mount Kurodake (Figure 1b). The section was developed on a gently undulated pyroclastic flow deposit that erupted approximately 34,000 years ago [51]. The trail section has an elevation difference of 9 m, from 1889.0 m a.s.l. to 1898.0 m a.s.l. The average slope is gentle at 4.3 degrees. The maximum depth of trail

erosion attains 1.5 m. The width of the tread surface of the trails, on which hikers/trekkers walk, ranges from 1.5 to 3.5 m.

A black-and-white aerial photograph taken in 1986 shows trail erosion. Heavy rainfall occurred during typhoons in August 2016. As a result, the lateral slope of the trail collapsed, and the collapsed soil was washed away and deposited outside the trail in the lower section, which was covered by alpine vegetation (Figures 2a and 3).

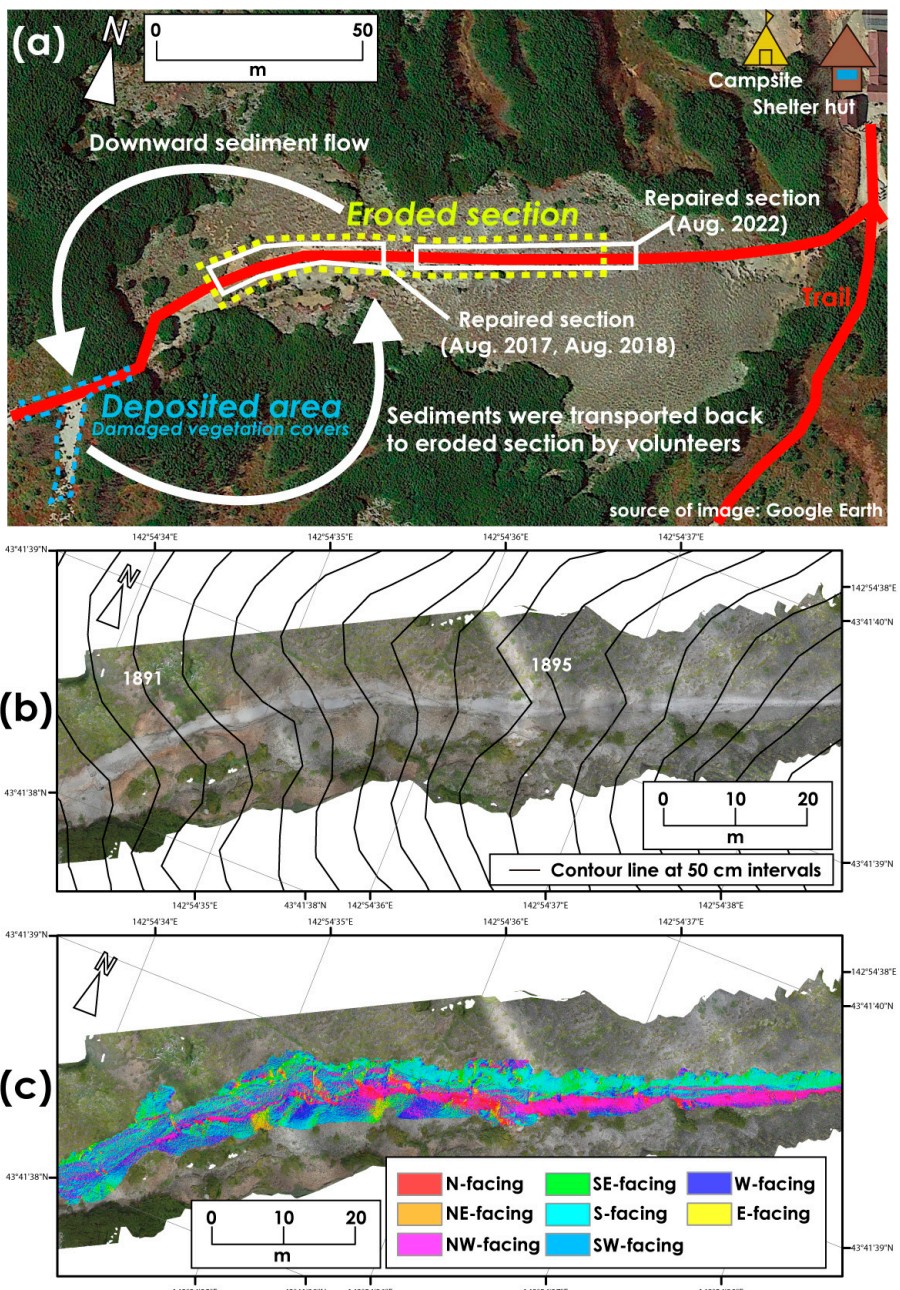

**Figure 2.** Detailed topography of the study area: (**a**) distribution of erosion and deposition on the studied trail section and location of repair works; (**b**) orthophotograph with contour lines prepared using pole photography on 5 August 2022; and (**c**) slope distribution on and around the trail surface. To show the surrounding topography and location of the trail, contour lines in (**b**) were created with a 10 m resolution DEM provided by the Geospatial Information Authority of Japan. Therefore, detailed erosion of the trail is not represented.

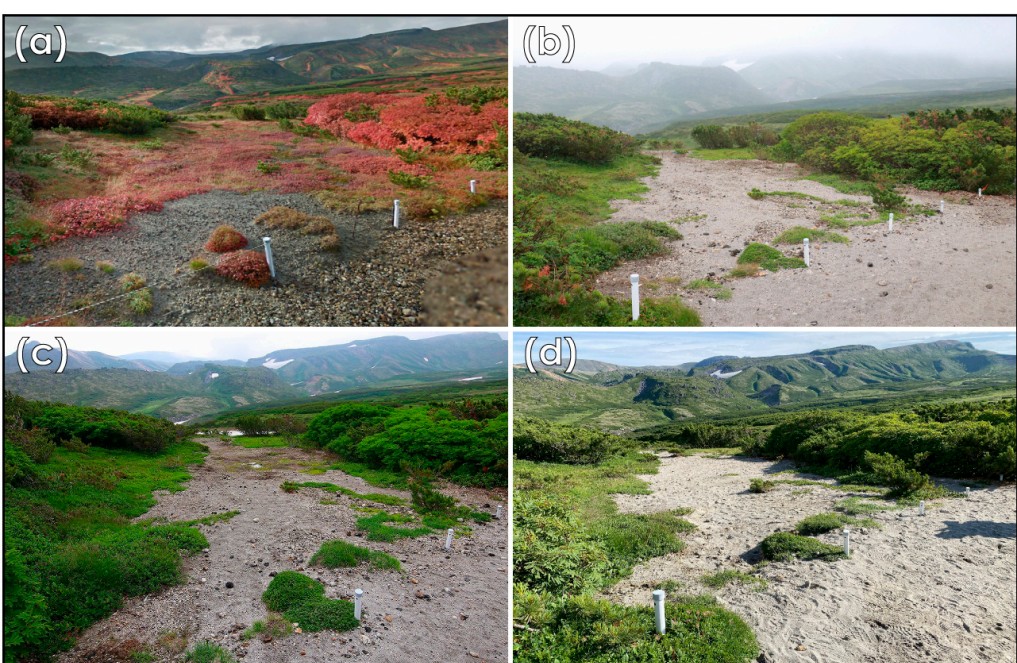

**Figure 3.** Deposition history of eroded soil flow on off-trail vegetation cover: (**a**) September 2014 (from Google Street view); (**b**) August 2018 (provided by *Yamamoritai*); (**c**) July 2019 (provided by *Yamamoritai*); and (**d**) August 2022 (provided by *Yamamoritai*).

### 2.2. Field Survey

2.2.1. The Use of Unmanned Aerial Vehicle and Pole Photography

In this study, a high-quality digital camera (RICOH GR) was attached to the DJI Phantom 2 in August 2016. An unmanned aerial vehicle (UAV) flew 8 m above the ground to obtain detailed topographic information regarding the ground surface. To counter the weaknesses of UAVs, such as battery issues in cold climates and remote areas, as well as a camera-shaking issue due to strong winds, this study adopted pole photography for all surveys from 2017 onwards (Figure 4a). RICOH GR2 was mounted on a 5 m long pole. The camera was operated using a smartphone via a Wi-Fi connection. Photographs were taken along both sides of each trail section from various angles, allowing an overlap of more than 80% of the consecutive photographs. Photographs were taken every year from September 2016 to August 2022, except in 2020, because of the spread of COVID-19. RICOH GR and GR2 have a large image sensor (APS-C) and weigh 251 g. They are suitable for attachment to Phantom 2 and a 5 m long pole for vertical photography.

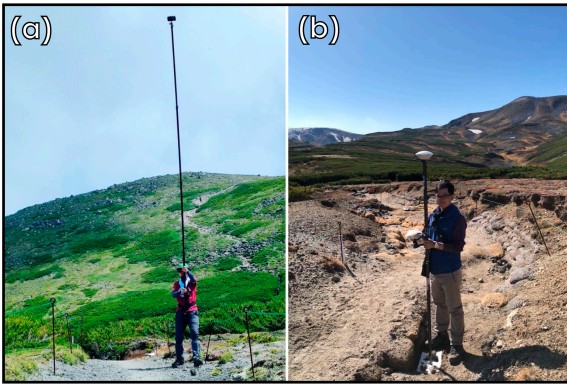

**Figure 4.** Two platforms were used to obtain data: (**a**) pole photography with RICOH GR2; and (**b**) GNSS receiver on the surveyed trail.

2.2.2. Precise Measurements of Coordinates and Elevation

Seven to fifteen ground control points were obtained on the trail by Post-Processing-Kinematic GNSS using Nikon-Trimble Geo 7X and Zephyr Model 2 antenna (Figure 4b) for the precise georeferencing of digital elevation models (DEMs) and orthoimages (i.e., UTM zone 54 N with the data from WGS84). Because there were no obstacles, such as tall trees, near the GNSS antenna, the number of satellites acquired at any given time ranged from 28 to 32. A single ground control point measurement took 3–5 min. The post-processing was performed with Trimble Pathfinder Office software [61], using the observed data and electronic data points provided by the Geospatial Information Authority of Japan [62]. A horizontal accuracy of 1 cm and vertical accuracy of 2 cm were obtained.

*2.3. Image Analysis*

2.3.1. Generation of High-Resolution DEMs and Orthoimages

Structure-from-motion and multi-view-stereo (SfM-MVS) photogrammetry was employed as the method of image processing in this study. This photogrammetry allows a three-dimensional interpretation of the topographic changes to the trail surface and measurement of the volume of erosion and deposition (Table 1). Agisoft Metashape was used for the structure-from-motion and multi-view-stereo analysis of the acquired image data for matching images and reconstructing the trail surface. This image processing enabled the creation of point cloud, 0.01 m resolution DEMs, and 0.05 m resolution orthoimages of the trail. A study conducted in Appalachia used 1 m resolution data generated using LiDAR [32]. This is insufficient if applied to the understanding of small phenomena, such as is the case in this study. An ultra-high-resolution DEM, such as a 1 cm resolution DEM, will succeed in detecting small changes. It is important to create data with a resolution appropriate for the extent and purpose of the study area.

**Table 1.** Summary of survey history, number of data, and DEM accuracy.

| Survey Date | No. of Images | No. of Point Clouds | RMSE of Check Points (m) |
|---|---|---|---|
| 20 September 2016 | 417 | 16,462,905 | 0.019 |
| 26 July 2017 | 384 | 34,514,188 | 0.012 |
| 29 July 2017 | 243 | 30,421,804 | 0.015 |
| 7 August 2018 | 647 | 60,152,438 | 0.017 |
| 30 August 2019 | 621 | 37,848,194 | 0.028 |
| 3 September 2021 | 736 | 52,399,616 | 0.024 |
| 5 August 2022 | 1129 | 63,917,295 | 0.015 |

Half of the ground control points were used as checkpoints to assess the accuracy of the DEMs. Checkpoint data were excluded when creating the DEMs.

2.3.2. Trail Surface Changes over the Study Period

The difference in DEMs (DoD), which is the subtraction of the previous year's DEM from the DEM of the following year, was used to understand the volume of erosion and deposition for the period and to detect the eroded and deposited areas of the repaired trail.

DEMs produced from structure-from-motion and multi-view stereo usually contain uncertainties and errors in the coordinates and elevation owing to the ground-control point precision and image resolution. Therefore, the simple subtraction of DEMs without accounting for uncertainties and errors did not result in significant changes [63,64]. To estimate the probability that trail surface changes were predicted by DoD, a 95% confidence interval was used as the threshold in this study. The DoD analysis employed Geomorphic Change Detection software [63], which is an add-on for the ESRI Arc Map 10.8.1. The geomorphic-change-detection method has been used in many studies to reveal land surface changes using DEMs [65]. It has also been used in studies on recreational ecology, including mountain trail studies [23] and campsite studies [29]. This analysis excluded areas where

repair works had been applied and areas with vegetation cover, to show the natural elevation changes in topography.

Furthermore, the ground surface before the emergence of erosion when the trail was formed was reconstructed by extrapolating from current contour lines. Then, a triangulated irregular network and 1 cm resolution DEMs were created, and the long-term changes in the trail surface were discussed. The long-term erosion volume from the initial stage of trail development was estimated by subtracting the current DEM from the reconstructed DEM.

### 2.4. Repair Works

Systematic trail management in Daisetsuzan National Park with volunteers began in the 2010s, and there has been no study of trail management or repairs since the trails were created. Minor repair work was conducted by the Ministry of the Environment before 2016. Wood steps and sandbags were installed on the tread surface to stop sediment movement. However, these construction projects did not work sufficiently against the heavy rainfall in August 2016.

Repair work on the eroded trails was performed in August 2017, August 2018, and August 2022. *Yamamoritai*, which is a small private entity that maintains the trails in Daisetsuzan National Park, the Ministry of the Environment Hokkaido Regional Environment Office, and Hokkaido's Kamikawa Sub-Prefectural Government, recruited about 30–60 hikers/trekkers as volunteers to help with the repair work. The authors participated as volunteers in this repair work.

### 2.4.1. Adopted Repair Work Methods to the Trail Section

There were two main methods of repair applied to the trail sections: (1) soil-filled palm-fiber bags and (2) wood steps. The soil-filled palm-fiber bags were installed for two purposes: (1) to prevent the soil from being washed away by surface water from the lateral slopes (Figure 5a), and (2) to hold soil so as not to prevent galley erosion on the tread surface (Figure 5b). The first purpose of the installation was to minimize further lateral erosion, that is, trail widening. The second purpose was to minimize the vertical erosion of the tread. For this purpose, existing wood steps were also utilized at some sites along with soil-filled palm-fiber bags (see the center of Figure 5a). The total number of soil-filled palm-fiber bags used was 300 in 2017 and 218 in 2018. In addition, some bags were cut to make mats, which were placed on the base of the lateral sloes for a 60 m section in 2017 and a 47 m section in 2018.

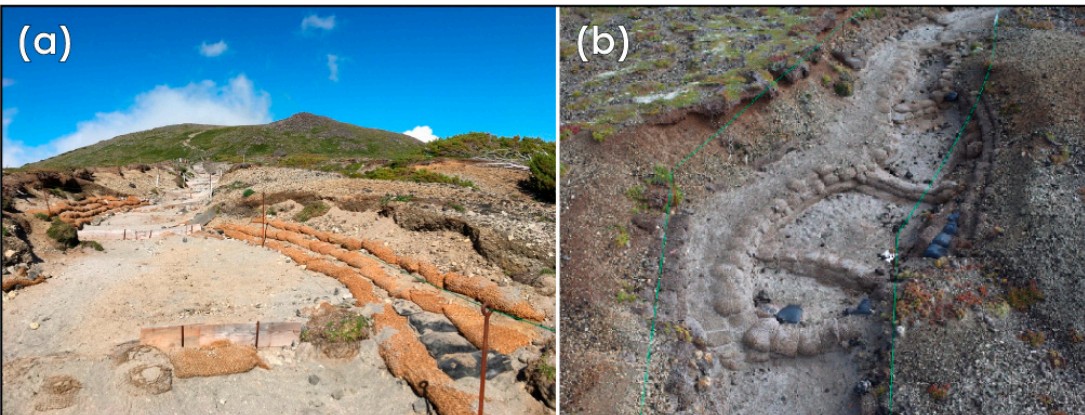

**Figure 5.** Soil-retaining structures for two purposes: (**a**) soil-filled palm-fiber bags on lateral slopes in 2017; and (**b**) soil-filled palm-fiber bags on tread surfaces in 2018. Note that the palm-fiber bags along with the wood steps were used on the tread surface in (**a**), but the wood steps were removed, as shown in (**b**).

2.4.2. Repair Works Applied to the Trail Section

The eroded soil flowed 50–60 m downward and covered the alpine vegetation, as previously stated (Figures 2 and 3). The first repair in 2017 involved removing the sediment from the deposited area (Figures 2a and 6a) and transporting it back to the eroded section (Figures 2a and 6b). The volunteers were briefed by a technical specialist belonging to *Yamamoritai* before work began. The technical specialist not only explained the workflow, but also explained why serious erosion occurred in Daisetsuzan National Park. The technical specialists emphasized the importance of observing the volunteers. The volunteers were divided into three teams based on their roles. The first team packed the sediment into palm-fiber bags (Figure 6a). Attention was paid to the alpine vegetation buried by the sediment flowing from the upper section of the trail and not to include the vegetation under the sediment in the palm-fiber bags. The second team transported the palm bags filled with soil to the upper section (Figure 6b), and the third team placed the soil-filled palm-fiber bags in the upper section (Figure 5). Volunteers from the trailhead carried out the palm-fiber bags used for the repairs. Palm-fiber bags are lightweight and easy to carry. Because the site is in the special protection zone of the national park (Figure 1a), as well as in the Daisetsuzan Grade 3 section, only the minimum necessary repairs were performed to protect the pristine natural landscape. Therefore, no materials from the outside were used, except for the natural palm-fiber bags.

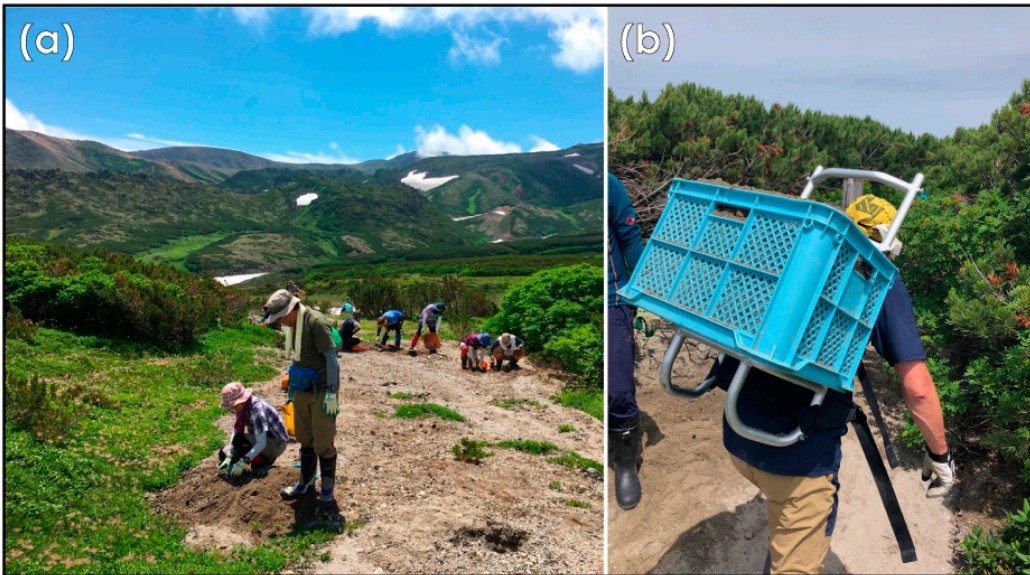

**Figure 6.** Involvement of volunteers: (**a**) volunteers packed the sediment into palm-fiber bags in the sedimented area and (**b**) volunteers transported the soil-filled palm bags to the upper section in the plastic case.

## 3. Results

### 3.1. Repair Works

The lateral slopes of the trail collapsed owing to massive erosion caused by heavy rainfall in August 2016, the year before the first repair. Consequently, the slope degrees of both sides became nearly vertical and highly unstable (Figure 7a). Palm-fiber bags were placed on the base of the lateral slopes to hold the collapsed soil from the top of the lateral slopes (Figures 7b and 8a).

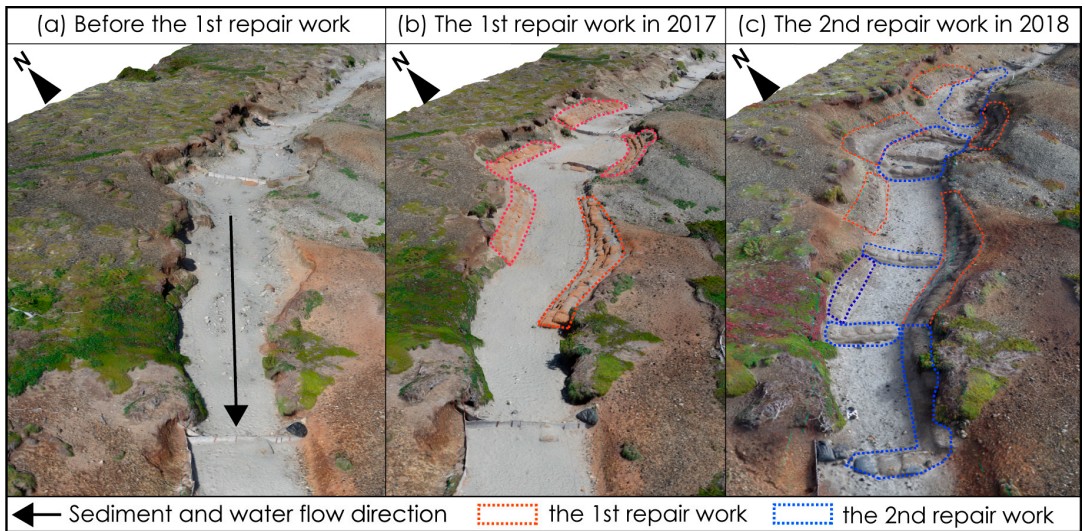

**Figure 7.** Trail surface (**a**) before the 1st repair work in 2017; (**b**) after the 1st repair work in 2017; and (**c**) after the 2nd repair work in 2018 (prepared using the UAV and pole-photography mappings).

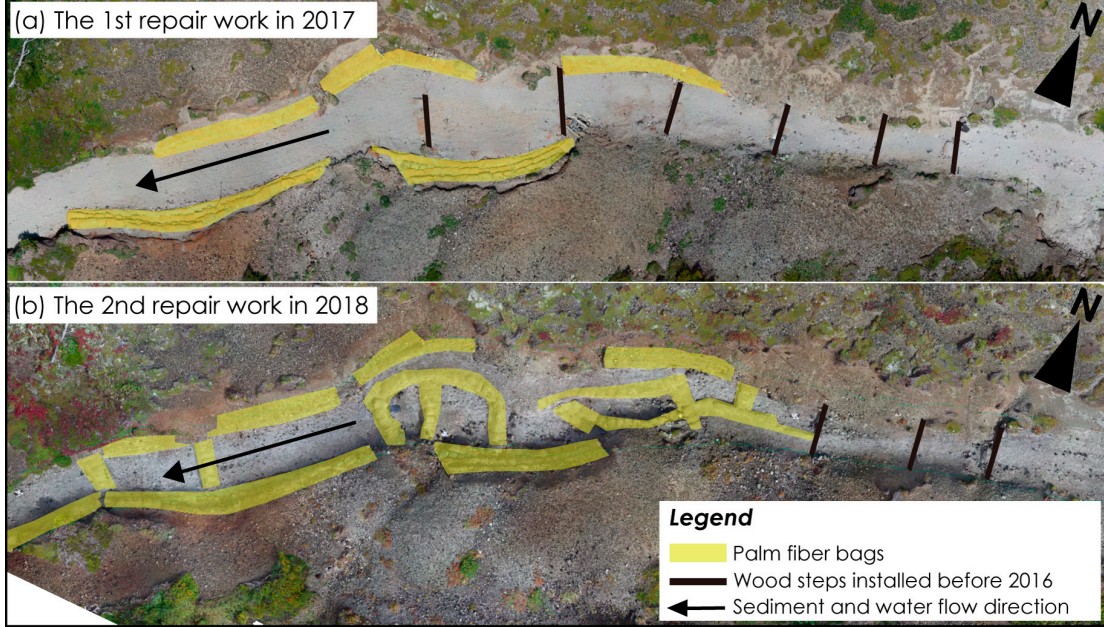

**Figure 8.** Trail surface (**a**) before the 1st repair work in 2017; and (**b**) after the 1st repair work in 2017 (prepared using the UAV and pole-photography mappings).

The second repair work was performed in August 2018 (Figure 7c). More erosion occurred in the upper section than in the section repaired during the first repair, and the eroded sediment moved downward. Therefore, the installation of palm-fiber bags for lateral slope protection was expanded, and wood step structures were installed on the tread surface. Palm-fiber bags were used for five earth retentions (Figures 7c and 8b). Careful attention was paid to the height of the earth-retaining structure when installing it, as a large protrusion of the earth-retaining structure from the ground could lead to further vertical erosion owing to falling surface water. In addition, it was important to design the steps of the earth-retaining wall, because it is difficult for elderly hikers/trekkers to lift their legs high while walking.

The third repair project was conducted in August 2022. From 2017 to 2022, vertical erosion was observed in the unrepaired areas. The sediments flowed directly downstream of the trail or the trailside.

*3.2. Evaluation of the Repair Works Based on Trail Surface Changes*

It was assumed that the trail surface had no erosion during the initial stage, when the trail was formed in 1923. By creating a supposed triangulated irregular network and DEM for the initial stage in 1923, and by subtracting the DEM in 2022 from the DEM in 1923, the long-term erosion was estimated to be $431.05 \pm 17.60$ m$^3$. A simple calculation indicated that $4.89 \pm 0.2$ m$^3$ of erosion had occurred annually.

After heavy rainfall in August 2016, the changes from September 2016 to July 2017 showed the collapse of the lateral slopes of the trail and vertical erosion of the tread surface (Figure 9a). The collapsed sediment remained intact and was not transported downward. A slight but significant sediment discharge of $-0.83 \pm 1.02$ m$^3$ was observed (Table 2).

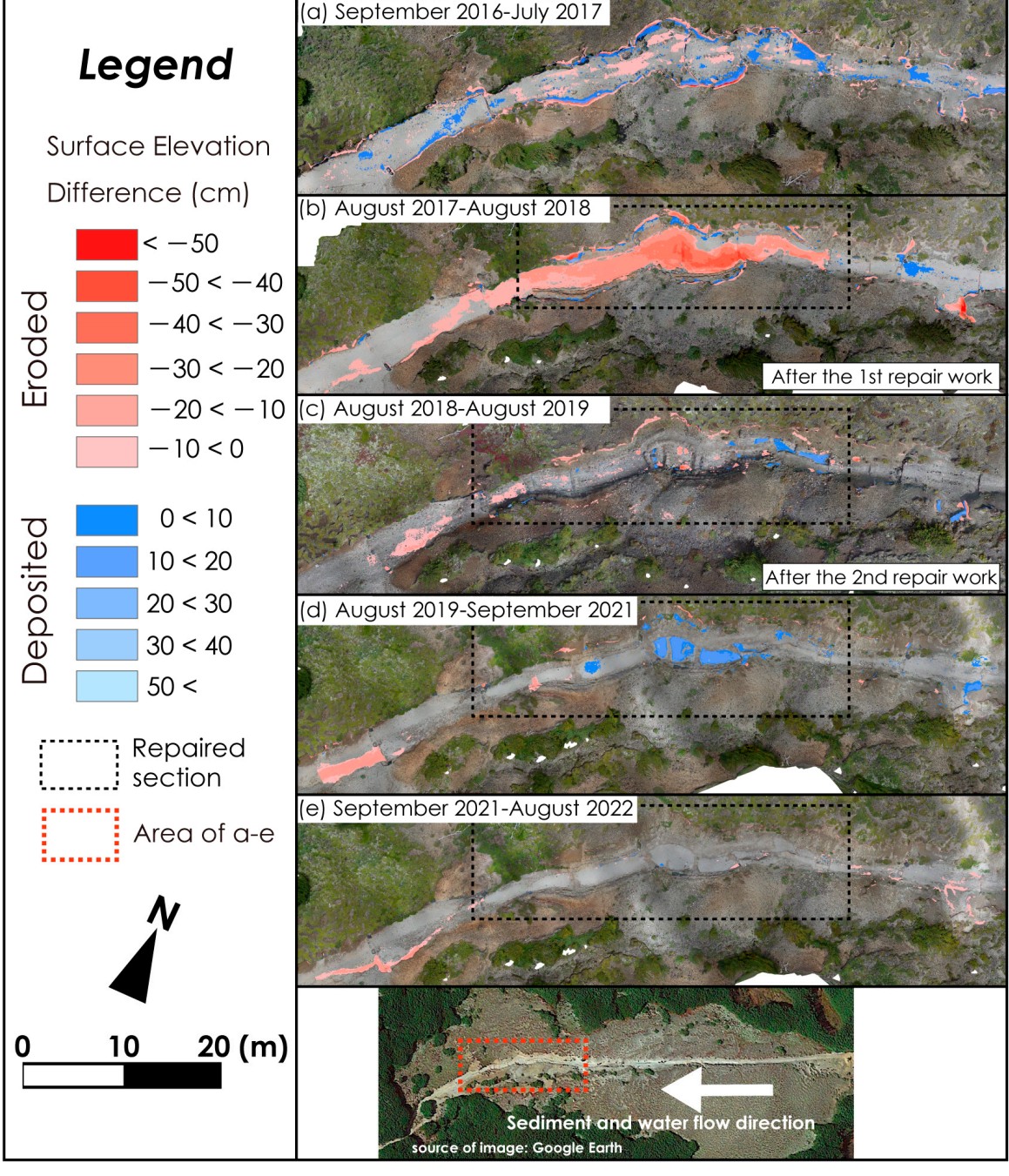

**Figure 9.** The trail surface changed from 2016 to 2022 in the section repaired in August 2016 and August 2017. See Figure 2a for the location of the section.

**Table 2.** Summary of surface change volume during the study period.

| Survey Period | Eroded Volume (m³) | Deposited Volume (m³) | Total Net Volume Difference (m³) |
|---|---|---|---|
| (a) 20 September 2016 to 26 July 2017 | 3.56 ± 0.75 | 2.73 ± 0.68 | −0.83 ± 1.02 |
| (b) 29 July 2017 to 7 August 2018 | 15.23 ± 2.52 | 0.52 ± 0.17 | −14.71 ± 2.52 |
| (c) 7 August 2018 to 30 August 2019 | 2.39 ± 0.87 | 1.01 ± 0.28 | −1.38 ± 0.92 |
| (d) 30 August 2019 to 3 September 2021 | 2.03 ± 0.63 | 1.95 ± 0.62 | −0.07 ± 0.89 |
| (e) 3 September 2021 to 5 August 2022 | 3.27 ± 1.18 | 0.14 ± 0.03 | −3.12 ± 1.18 |

Vertical erosion occurred at many sites between August 2017 and August 2018 (Figures 9b and 10b). Further collapses continued on the south-facing slopes of the trail; however, the collapsed soil mass was trapped by the soil-filled palm-fiber bags (Figures 9b and 11c–e). However, less collapse occurred on the north-facing lateral slope of the trail. Vertical erosion was observed across the entire trail tread. The red line in the profile in Figure 11c suggests 45 cm of vertical erosion, resulting in a large amount of sediment discharge, i.e., −14.71 ± 2.52 m³. In addition, surface water damaged the soil-filled palm-fiber bags at the base of the lateral slopes. However, no lateral slope collapse was observed; therefore, no widening of the trail occurred.

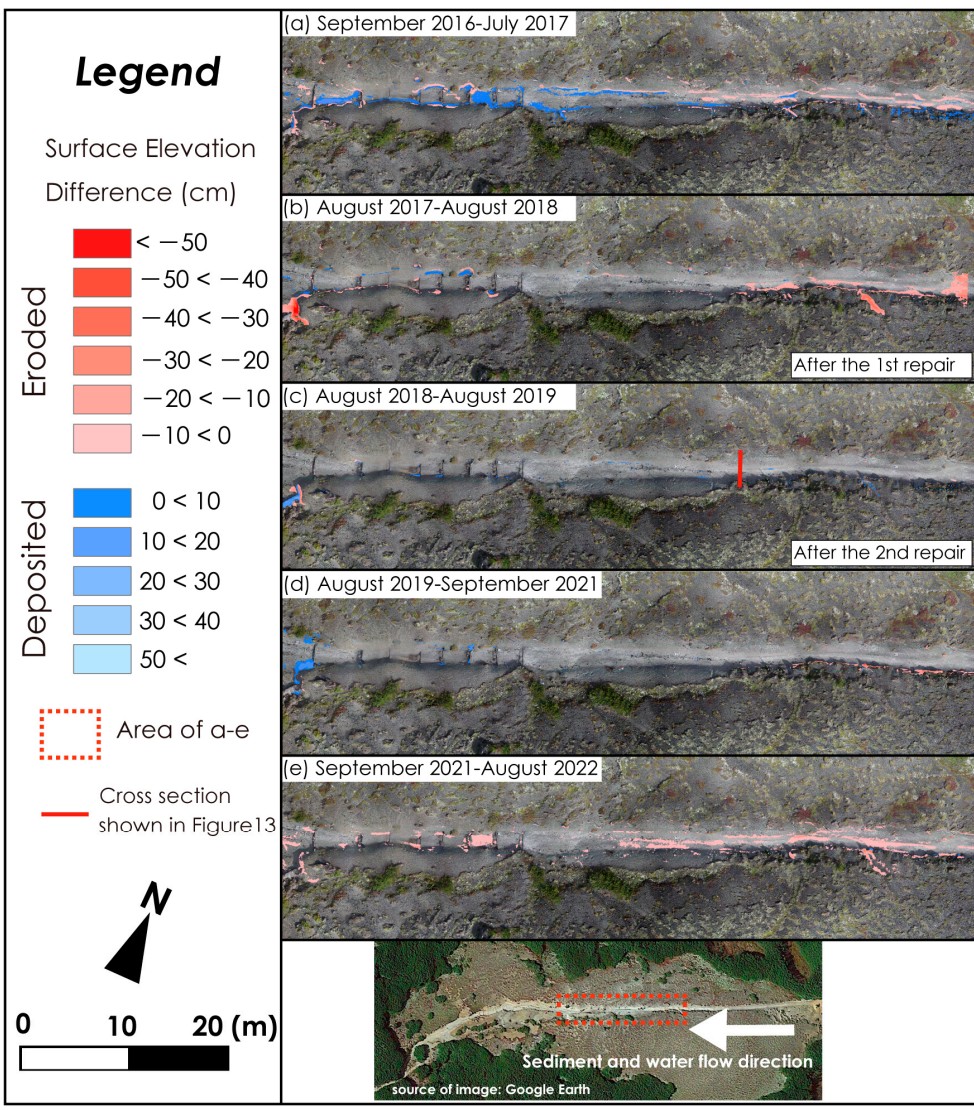

**Figure 10.** The trail surface changed from 2016 to 2022 in the upper section repaired in August 2022. See Figure 2a for the location of the section.

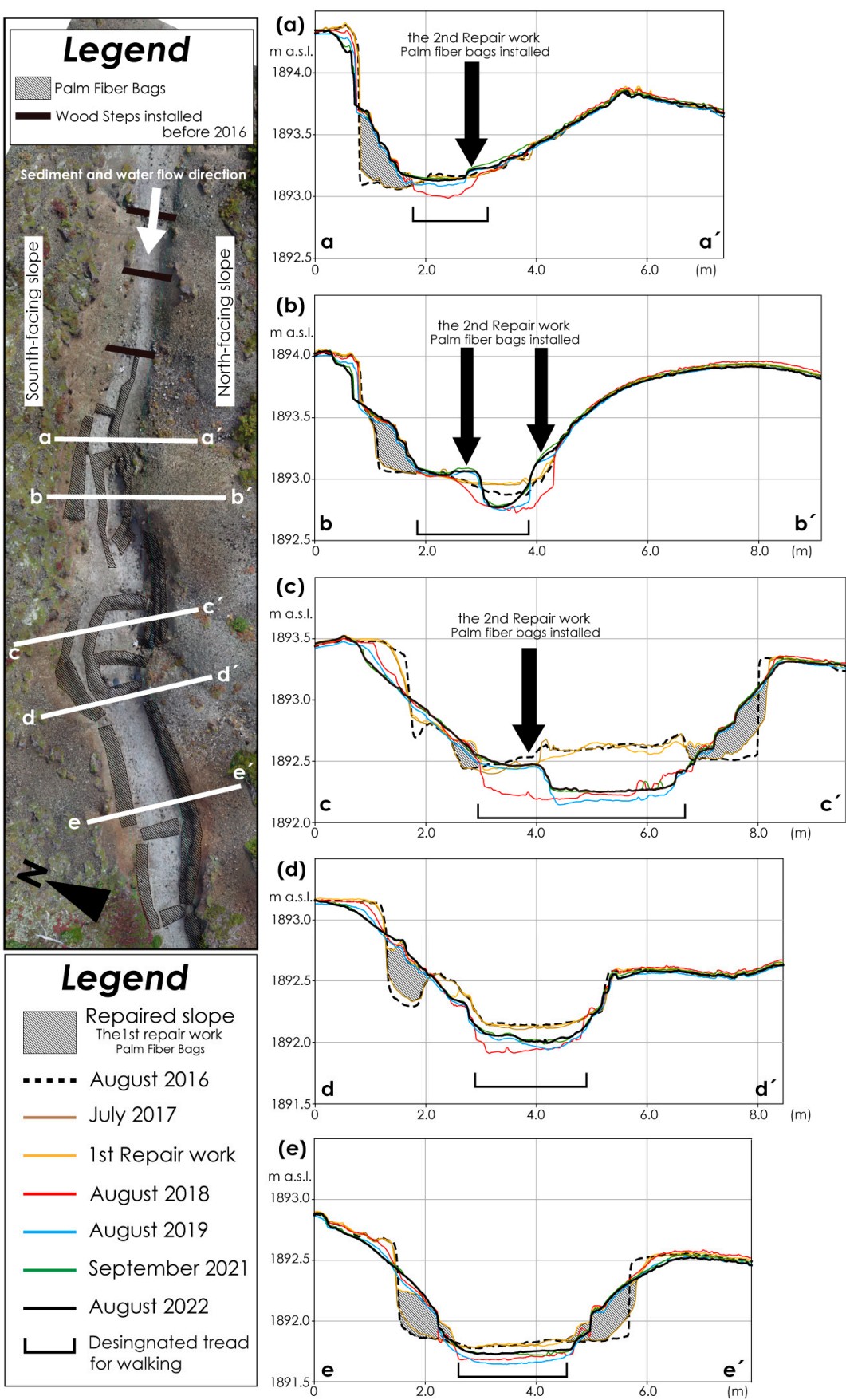

**Figure 11.** Trail profile changes at (**a–e**) in the repaired section from 2016 to 2022.

From 2018 to 2019, erosion was still visible on the south-facing lateral slopes. The destination of the eroded sediment was unclear, because the second repair work was conducted during this period. However, Figure 9c suggests that the deposition occurred at the base of the lateral slope. Figure 7c shows the anthropogenic influence of the second repair work.

From August 2019 to September 2021, deposition occurred at locations where earthen berms were installed. There was continued erosion on the south-facing slope of the trail; however, sediment was received by the soil-filled palm-fiber bags (Figure 9d). No significant erosion was observed in the upper segment.

From September 2021 to August 2022, some parts of the trail were eroded in the upper segment. No erosion or sedimentation was observed in the repaired segment (Figures 9e and 10e). However, erosion was noticeable in the upper section, and a sediment discharge of $-3.12 \pm 1.18$ m$^3$ was observed.

More changes were observed in the upper part of the lateral slopes on the south-facing slopes than on the north-facing slopes (Figure 9a,b,d). The soil fell from the top of the lateral slopes, with the palm-fiber bags preventing the sidewalls from becoming vertical.

## 4. Discussion

### 4.1. Effectiveness of Repair Works

This study reveals the effectiveness of the protection and stabilization of the lateral slopes five years after the first repair work in 2017. The second repair work in 2018 achieved the objective of storing sediment until 2021; however, further soil storage was no longer possible by 2022.

### 4.1.1. Successful Achievements

Three heavy precipitation events occurred in the study area after the first repair in August 2017 (Figure 12). As stated earlier, the red line shown in the profile in Figure 9c suggests 45 cm of vertical erosion and $-14.71 \pm 2.52$ m$^3$ of sediment discharge was likely due to the heavy precipitation events. The installation of palm-fiber bags on the base of the lateral slopes successfully protected the soil from surface water. Without this protection in 2017, further loss of collapsed soil from the lateral slopes could have occurred during the heavy rain in 2018.

Continuous soil movement occurred on the south-facing lateral slope of the trail, even after the first repair work. As stated earlier, the south-facing slope was more easily eroded by collapse than the north-facing slope was.

Protecting the base of the lateral slopes with palm-fiber bags prevented erosion by surface water and the further widening of the trail. In addition, by installing palm-fiber bags on the lateral slopes, the vertical slopes were transformed into gentle slopes by holding soil from the top of the lateral slopes. Moreover, stabilizing the movement of sediment on a lateral slope is expected to promote vegetation recovery and further stabilize the slopes.

From August 2019 to September 2021, an accumulation of 15–20 cm was observed in the soil-filled palm-fiber bags. No erosion occurred in the upper section, resulting in a large amount of deposition relative to the amount of erosion. The large increase in deposition might be due to the inflow of sediment from the slopes surrounding the trail or piping.

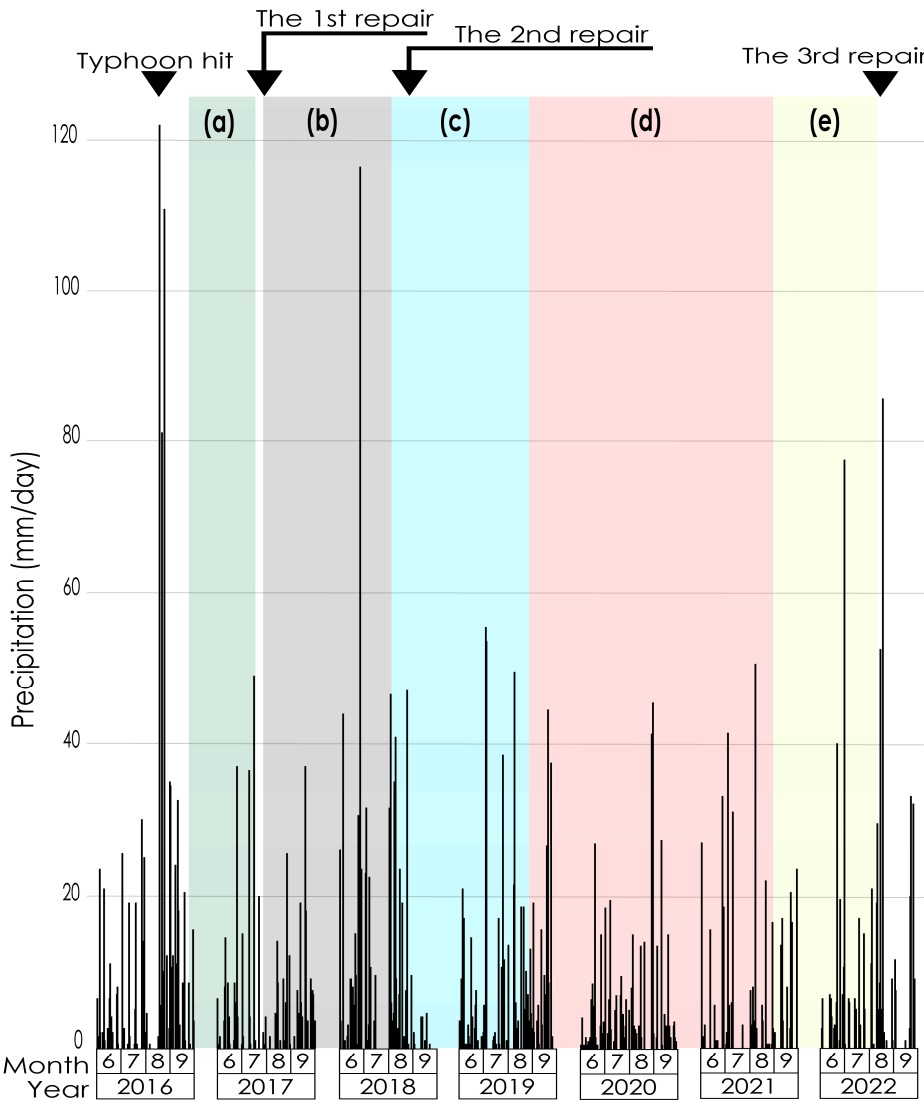

**Figure 12.** Precipitation (mm/day) at the Sounkyo Meteorological Station in the summer during the study period. Letters (**a–e**) correspond to those in Figures 9 and 10, and Table 2. Precipitation data from the Meteorological Agency [59]. See Figure 1a for the location of the Sounkyo Meteorological Station.

4.1.2. Areas in Need of Improvement

The overall depth of the eroded trail showed a large negative value from 2021 to 2022. This means that the first-eroded and then temporarily deposited soil flowed outside the survey section, that is, into the lower part of the trail and the vegetated off-trail area.

Sediment storage in the earth-retaining structures became full in 2019–2021; hence, the sediment had to be returned to the upper section by volunteers. Even without heavy rainfall events, such as that of August 2016, this storage system would not work without regular maintenance. Frequent and regular maintenance is necessary to transport the overflowing sediment back to the upper section.

Trampling on the earth-retaining structure by hikers/trekkers was naturally observed. Although palm-fiber bags last approximately 10 years, they need to be replaced periodically because of the damage caused by the trampling of hikers/trekkers.

Repair works by *Yamamoritai* began in the 2010s, and priority is now being given to areas where the degradation is most severe. Figure 1 shows the areas that were repaired by *Yamamoritai* up to today. However, it will be necessary to prioritize the repair sections based on continuous monitoring soon.

### 4.2. Future Trail Management in Daisetsuzan National Park

Figures 9–12 suggest that significant erosion occurs after heavy rainfall events above 60 mm day$^{-1}$ (e.g., 2017 and 2018). Rainwater easily penetrates the ground because of the thick accumulation of volcanic materials. However, some water cannot permeate because of the existence of a seasonally frozen layer in spring to late summer that serves as an impermeable layer [66,67]. Consequently, the oversaturated water flows on the trail surface. Owing to the harsh climatic conditions that produce the periglacial terrain, the organic soil is very thin. The volcanic sandy soils are fragile, and the freely meandering surface water has led to the collapse of the lateral slopes of the trails. This condition is the reason why trail erosion is more severe in Daisetsuzan National Park than elsewhere found in previous studies. The author of ref. [15] found that the occurrence of erosion is highly dependent on soil characteristics when the intensity of use is the same. Therefore, the number of hikers/trekkers alone cannot explain the reason for the trail erosion when compared to other national parks with a similar numbers of hikers/trekkers.

A study in Alaska [43] raised concerns regarding trail erosion due to permafrost thawing. The existence of permafrost is also known in Daisetsuzan National Park [56–58,68]. However, the exact depth and thickness of the permafrost along the trails are not well known. The subsidence caused by the thawing of permafrost leads to significant trail erosion. In addition, gelifluction occurs in the oversaturated active layer of the permafrost and even on the seasonally frozen layer from spring to late summer, resulting in trail erosion. As a study in Taiwan [18] showed, sounding underground at the point where the trail is located is essential for predicting future erosion.

The repair work transported a large amount of sediment to the upper section to rehabilitate the alpine vegetation covered by sediment. However, the erosion accumulation balance was still negative and further inflow into the vegetation-covered area was possible. This suggests that if the current trail remains in use, it must be repaired and managed to control the movement of eroded soil by surface water and freeze–thaw processes until the unconsolidated volcanic materials completely disappear. Compared to the results of [5], Daisetsuzan National Park has very rapid erosion rates [42], and drastic measures are required to secure sustainable trails. However, replacing trails is uncommon in Japan's mountain national parks, because national park authorities tend to avoid unprecedent undertakings. Further difficulties are expected because most of the trails are designated in the special protection zone in Daisetsuzan National Park.

Trail closure is an option for addressing further soil erosion. However, even if the trail is replaced during closure, it is difficult to completely prevent erosion because of the surface materials of Daisetsuzan National Park. In addition, sediment transport continued in the old trail after closure. Continuous trail erosion was observed on the old trail, which was no longer in use after temporary replacement [69].

Palm-fiber bags are used to control runoff on the trail and protect vulnerable areas. Although this maintenance approach requires frequent repairs, certain effects are clear as previously stated, and it is desirable to sustain the current trail. The installation of soil-filled palm-fiber bags and/or wood steps provides clear instructions to trekkers on where to walk. To prevent the further widening of the trail, it is important to clearly indicate where the hikers/trekkers should walk.

The authors of ref. [42] and one of the authors of this study (TW) conducted cross-sectional area measurements in the upper section in 1990, 1997, and 2013. Their results suggested that the maximum erosion depth was 35 cm from 1923 to 1990 (0.52 mm y$^{-1}$) and 70 cm from 1990 to 2013 (15.2 mm y$^{-1}$). Because the cross-sectional area measurement conducted in this study shows a maximum erosion depth of 110 cm from 2013 to 2022 (44.4 mm y$^{-1}$), the erosion rate has increased at an accelerating rate (Figure 13). One reason for the recent increase in the erosion rate may be that only three repair works have been performed. This suggests that repair work is becoming increasingly important.

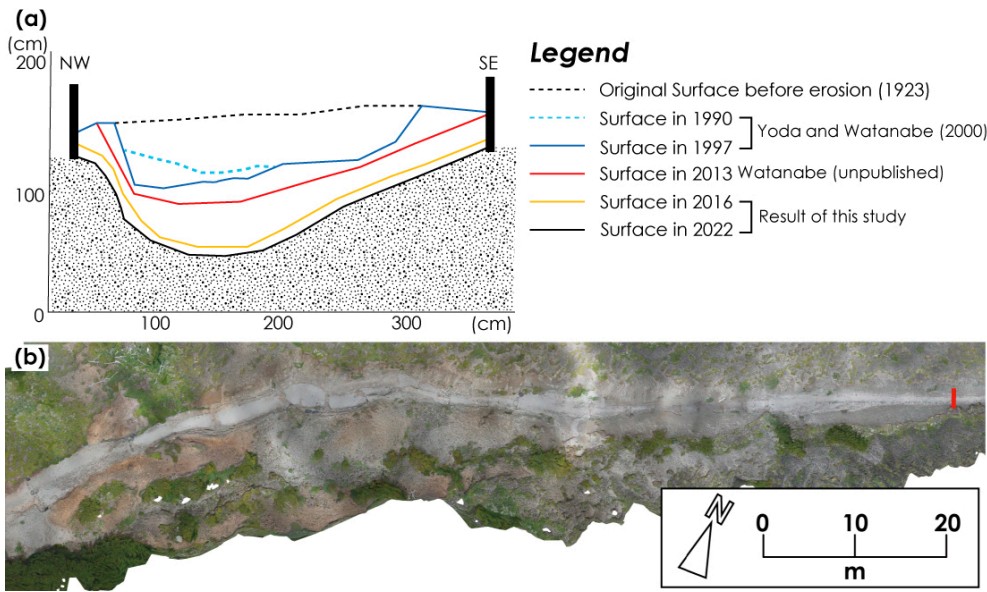

**Figure 13.** The long-term cross-sectional profile: (**a**) changes from 1923 to 2022; and (**b**) the location (red line) of cross section (**a**). See Figure 2b for the location of (**b**).

The management of the trail should include a combination of repair and monitoring. For example, it is important to remove fallen leaves and other debris frequently when draining a side-hill trail. The storage system does not function once sediment fills the earth-retaining structures. In Japan, emphasis has been placed on repair works, but regular monitoring is still insufficient and even lacking on many trails. The long-term recording of changes after repair works will lead to the evaluation of repair works and will be helpful for future repair works. Showing the effectiveness of repair work will enhance motivation for volunteer work.

In this study, the repaired sections were recorded once per year. Therefore, it was difficult to determine whether these changes occurred immediately after a heavy rainfall event. It may be necessary to record topographical changes for as many days as possible, including the days immediately before and after heavy rainfall. It may also be necessary to take photographs during freeze–thaw seasons. It is necessary to consider the survey timing for these different erosion processes in the future.

When quantitatively measuring trails with spatial coverage, as in this study, it is important to predict the section or area of possible trail impact. In the short periods of 2017 and 2018, repair works were performed by volunteer hikers/trekkers. Partnerships with hikers/trekkers are expected to play an important role in monitoring trails as well.

In addition, it will be desirable to develop and incorporate a nomogram of the suggested timing and approaches considered in this study, and how these differ from other parts of the world, which should be addressed in future studies. Monitoring trails with high frequency and covering a wide area is difficult because of the limited number of park managers and researchers. The provision of photographs and smartphone LiDAR images by volunteers helps fill the data shortage. Academics, along with the Ministry of the Environment, are currently preparing for such a data-accumulation framework in Daisetsuzan National Park.

## 5. Conclusions

This study used structure-from-motion and multi-view-stereo analysis to determine the effectiveness of trail repair works with the participation of volunteers, mainly composed of trekkers/hikers, based on long-term monitoring of the 130 m section near Mount Kurodake, Daisetsuzan National Park. It was confirmed that the palm-fiber bags stopped further erosion and sediment movement on the side slopes and tread surfaces of the eroded

trail. This study suggests that additional maintenance work is not required for approximately three years after repair work. Therefore, regular observations and quick decision making regarding further repair work are important for augmenting the sustainability of the trail. The results of this study will be an important basis for trail management in other trail sections of Daisetsuzan National Park, as well as other mountainous protected areas worldwide.

Erosion has been observed in the majority of the 300 km long trails in Daisetsuzan National Park. Mapping the distribution of erosion rates is required to determine which sections of the trails that experience erosion in the park should be prioritized for repair work. Unmanned aerial vehicle (UAV) surveys are indispensable for this purpose.

The soil erosion rate calculated using the maximum erosion depth increased from 0.52 mm y$^{-1}$ (1923–1990) to 44.4 mm y$^{-1}$ (2013–2022), suggesting that both frequent observations and repeat repair works are required. Trail maintenance through a combination of monitoring and repair work is vital, and the role of hikers/trekkers is becoming increasingly important. A nomogram of the local rates is important to increase the visibility and impact of this study.

Continuous repair works on trails with the participation of hikers/trekkers will require efforts by researchers to visualize the effects of such repair works.

**Author Contributions:** Conceptualization, T.W.; methodology, Y.K. and T.W.; software, Y.K.; validation, Y.K. and T.W.; formal analysis and investigation, Y.K. and T.W.; data curation, Y.K.; writing—original draft preparation, Y.K. and T.W.; writing—review and editing, Y.K. and T.W.; visualization, Y.K. and T.W.; supervision, project administration, and funding acquisition, T.W. All authors have read and agreed to the published version of the manuscript.

**Funding:** This research was partly funded by JSPS Kakenhi Research Fund (Grant-in-Aid), grant number 15K12451 (T.W.).

**Acknowledgments:** We sincerely thank the shelter hut managers and Rinyu Kanko for helping to store the survey equipment, the *Yamamoritai* for providing valuable trail photos, and all the trekkers who cooperated with us during the survey. Permits for the in situ research and drone flight were granted by the Ministry of the Environment and the Hokkaido Regional Forest Office of Forestry Agency.

**Conflicts of Interest:** The authors declare no conflict of interest.

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
