# Peer review of "Evaluation of the Effectiveness of Trail Repair Works Based on Three-Dimensional Monitoring around Mount Kurodake, Daisetsuzan National Park, Japan"

_sustainability, doi:10.3390/su151712794_

Round 1

Reviewer 1 Report

I recommend authors to read carefully the attached file and to consider all suggestions and corrections made.

English is really fine, so, tha authors only should correct the few, tiny mistakes that I suggest them to fix.

Author Response

Dear Reviewer 1

Reviewer 2 Report

This manuscript evaluates the effectiveness of repair works at 133m out of 300km of trails around Mount Kurodake, Daisetsuzan National Park in Japan based on three-dimensional monitoring. Trail science has focused primarily on quantitative measurements of erosion, with little attention paid to the destination of the eroded sediment. The added value of this work is the scientific confirmation of the effectiveness of the repair methods utilised. This paper can be considered for publication after minor changes taken into consideration the following sections:

Ln. 21-22; “Trail maintenance through a combination of monitoring and repair work is vital, and the role of hikers/trekkers is becoming increasingly important”. It will be desirable to develop and incorporate at this work (and summarized at the abstract) a nomogram of the suggested timing and approaches considered and how these differ from other parts of the world. Also, it should be considered if this approach is archeologically adopted by ancient inhabitants. If there are no such evidence it should specifically mentioned in the text,

Ln. 39-57; Although the authors are aware about the efforts in the international literature such as the Scottish Hill Paths, the trails at Grövelsjön, Rocky Mountain National Park, the Shei-Pa National Park, Taiwan, the Acadia National Park, the Torres del Paine National Park at Chile and at the "El Portalet" mountain trails in the Eastern Iberian Peninsula,

these works are hardly reviewed and the conclusions reached are not analysed. These lines should be expanded in order to explain in detail the worldwide relevance of the experiences accounted. In particular should be mentioned the trails or paths examined.

Ln. 85-89; “…studies of trails in high-elevation and/or cold climates similar to DNP have been conducted in the U.S., Iceland, Nepal, and Peru [49-54] …” ,

yet the impact on trails with similar number of visitors are the Bushland and track and restoration words in Australia and similar orography and climate the alpine environment of the gorge of Samaria in Greece where the trail erosion rates should be similar although the restoration approaches might furnish different outcomes.  

Ln. 131; “…in 1985, was - 3.8°C [57]. The area above 1,850 m is widely covered by periglacial landforms such as sorted polygons…”

For such alpine sites, not only the mean temperature but the range of max and min is necessary as well the days with snow cover or rain per year

Ln. 156; Fig 1a and 1b the area with rope way is not easy to identify.
A different thickness and colour for the line might help.

Ln. 190-191; “…using the observed data and electronic data points provided by the Geospatial Information Authority of Japan”

Suitable Refence(s) and web link(s) are essential to support this work.

Ln. 377-378; At fig.11 instead of lines with different colours the authors should use dashed lines with symbols and different widths as well as colours should be used. This will make the differences between the results visible between 2016 and 2022.

Ln. 379-383; Fig. 12 the daily rain is used but for erosion damages most relevant will be the rain rate.  

Ln. 381-382; The web page of the portal with the data must be included in the legend and the appropriate references should be incorporated (even if they are in Japanese).

Ln.504-506; “…regarding further repair work are important to augment the sustainability of the trail. The results of this study will be an important base for trail management in other trail sections of DNP, as well as other mountain protected areas worldwide…”.

The authors should explain further and perhaps indicate how they plan to select the subsequent sections for repairs. Now only 133m out of 300km were considered.

Ln. 512-515; “.., the maximum erosion depth has increased from --(2013–2022), suggesting that both frequent observations and repeat repair works are required. Trail maintenance through a combination of monitoring and repair work is vital…” a nomogram of the local rates should be of importance for increasing the visibility and the impact of this work.

The quality of English and grammar is well presented. Yet, I would like to recommend to the authors to avoid all acronyms especially at the conclusions and the abstract because these sections are read all scientific audience. This is valid even for acronyms that appear to be well established in modern literature. 

The quality of English and grammar is well presented. Yet, I would like to recommend to the authors to avoid all acronyms especially at the conclusions and the abstract because these sections are read all scientific audience. Also, there are several three-letter definitions such as DNP(17), MVS(9), MIC(5) and CSA(6) that are not essential in their introduction. This is valid even for acronyms that appear to be well established in modern literature.

Also, at line 519 the wording “conceptualization” should be changed.

Author Response

Dear Reviewer 2

Reviewer 3 Report

Dear Author,

This manuscript accurately be fictionalized. The aim of this manuscript is given shortly in the abstract section. The aim of the study in additionally mentioned in the section introduction. The section introduction was explained capably by references. The study area material and method were denoted precisely in the section of material and method. Results in detail were discussed with relative references.  Conclusions adequately were emphasized. The author's articles weren’t used as references in this manuscript. As a result, this manuscript is an original and recent study in my opinion. This manuscript is at the qualification that would be published in this journal.

Author Response

Dear Reviewer 3
